# Risk Factors, Prevalence, and Outcomes of Invasive Fungal Disease Post Hematopoietic Cell Transplantation and Cellular Therapies: A Retrospective Monocenter Real-Life Analysis

**DOI:** 10.3390/cancers15133529

**Published:** 2023-07-07

**Authors:** Eleni Gavriilaki, Panagiotis Dolgyras, Sotiria Dimou-Mpesikli, Aikaterini Poulopoulou, Paschalis Evangelidis, Nikolaos Evangelidis, Christos Demosthenous, Evangelia Zachrou, Panagiotis Siasios, Despina Mallouri, Anna Vardi, Zoi Bousiou, Alkistis Panteliadou, Ioannis Batsis, Marianna Masmanidou, Chrysavgi Lalayanni, Evangelia Yannaki, Damianos Sotiropoulos, Achilles Anagnostopoulos, Timoleon-Achilleas Vyzantiadis, Ioanna Sakellari

**Affiliations:** 1Bone Marrow Transplantation Unit, Haematology Department, G. Papanicolaou Hospital, 57010 Thessaloniki, Greece; panadolg@gmail.com (P.D.); mpesi12@hotmail.com (S.D.-M.); christosde@msn.com (C.D.); dmallouri@gmail.com (D.M.); anna_vardi@yahoo.com (A.V.); boussiou_z@hotmail.com (Z.B.); kirapanteliadou@gmail.com (A.P.); iobats@yahoo.gr (I.B.); mariannareti@gmail.com (M.M.); luizana6@gmail.com (C.L.); eyannaki@uw.edu (E.Y.); dsotiro@otenet.gr (D.S.); achanagh@gmail.com (A.A.); ioannamarilena@gmail.com (I.S.); 2Second Propedeutic Department of Internal Medicine, Hippocration Hospital, Aristotle University of Thessaloniki, 54642 Thessaloniki, Greece; pascevan@auth.gr (P.E.); evangeln@auth.gr (N.E.); 3Department of Microbiology, Medical School, Aristotle University of Thessaloniki, 54124 Thessaloniki, Greece; catherinepoulopoulou@gmail.com (A.P.); zachrouev@yahoo.gr (E.Z.); panossiassios@gmail.com (P.S.); tavyz@auth.gr (T.-A.V.)

**Keywords:** antifungal, CAR T cell therapy, hematopoietic cell transplantation (HCT), immunosuppression, invasive fungal infection, mold, prophylaxis, risk factors, yeast

## Abstract

**Simple Summary:**

Invasive fungal infections (IFD) are a significant cause of morbidity and mortality in patients with hematological malignant disorders. The aim of this retrospective study was to investigate the prevalence and outcome of IFD in patients who received cellular therapies. Adult hematopoietic cell transplantation (HCT), chimeric antigen receptor (CAR) T cell, and gene therapy recipients were retrospectively enrolled. We studied 950 patients who received cellular therapies. No CAR T cell and gene therapy patient showed IFD whereas 3/456 autologous HCT recipients who suffered from primary refractory/relapsed lymphomas presented with probable IFD. A total of 11/473 allogeneic HCT recipients developed probable IFD, possible IFD was found in 31/473 and IFD was proven in 10/473. Three patients were reported to have a second episode of IFD. IFDs were associated with poor outcomes in allogeneic HCT recipients. Further studies are needed to improve diagnostics and therapeutics in high-risk populations, such as allogeneic HCT recipients in whom IFD is associated with poor outcomes.

**Abstract:**

(1) Background: Autologous, allogeneic hematopoietic cell transplantation (HCT) and other cellular therapies, including CAR T cell and gene therapy, constitute a cornerstone in the management of various benign and malignant hematological disorders. Invasive fungal infections (IFD) remain a significant cause of morbidity and mortality in HCT recipients. Therefore, we investigated the prevalence and risk factors of IFD following HCT and other cellular therapies in an era of novel antifungal prophylaxis. (2) Methods: In this study, we retrospectively enrolled adult HCT recipients who were treated at our JACIE-accredited center according to standard operating procedures over the last decade (2013–2022). (3) Results: 950 patients who received cellular therapies were studied. None of the 19 CAR T cell and neither of the two gene therapy recipients developed IFD whereas 3/456 autologous HCT recipients who suffered from primary refractory/relapsed lymphomas presented with probable IFD. Overall, 11 patients who received allogeneic HCT experienced probable IFD, possible IFD was found in 31/473, and IFD was proven in 10/473. A second IFD episode was present in three patients. Four-year OS was significantly lower in proven compared to probable IFD (*p* = 0.041) and was independently associated with HCT-CI (*p* = 0.040) and chronic GVHD (*p* = 0.045). (4) Conclusions: In this real-world cohort, the prevalence of proven and probable IFD in an era of novel antifungal prophylaxis was found to be relatively low. However, IFDs were associated with poor outcomes for patients who received allogeneic HCT.

## 1. Introduction

Several studies have established the essential role of cellular therapy in various hematological diseases. Indeed, autologous, and allogeneic hematopoietic cell transplantation (HCT) and chimeric antigen receptor (CAR) T cell therapy remain a cornerstone in the management of various benign and malignant disorders. HCT has proven its curative potential in patients with many malignant and non-malignant hematological disorders, including leukemia, lymphomas, multiple myeloma, thalassemia, and aplastic anemia [1,2,3,4]. Interestingly, alloHCT has been used effectively in the treatment of various disorders, such as medication-related osteonecrosis of the jaw in experimental rat models [5]. CAR T cell therapy is considered a promising and emerging treatment for hematologic malignancies. In Europe, three CD19-targeted CAR T products have been approved for the treatment of lymphoid malignancies: tisagenlecleucel (tisa-cel) for the treatment of diffuse large B-cell lymphoma (DLBCL) and B-cell acute lymphoblastic leukemia (B-ALL), axicabtagene ciloleucel (axi-cel) for the treatment of DLBCL and primary mediastinal B-cell lymphoma (PMBCL), and brexucabtagene autoleucel for the treatment of mantle cell lymphoma (MCL) and B-ALL.

Infections represent a leading cause of life-threatening complications in the HCT setting related to several factors: (1) Net state of immunosuppression (e.g., acute myeloid leukemia [AML] or aplastic anemia, not in first complete remission, neutropenia, lymphopenia, graft failure, age > 40 years old, graft-versus-host disease [GVHD], immunosuppressive agents); (2) Graft characteristics (e.g., matched unrelated donor or mismatched related donor, T cell-depleted graft, CD34 < 2 × 10^6^/kg, umbilical cord blood); (3) conditioning regimen (myeloablative); (4) organ dysfunction, and (5) exposure to pathogens [6,7,8,9,10]. Data regarding the incidence of infections in patients receiving CAR T cell constructs are scarce; however, neutropenia, impaired cellular immunity, B cell aplasia, and hypogammaglobulinemia, along with the use of tocilizumab and/or corticosteroids for the treatment of cytokine release syndrome (CRS) and/or neurotoxicity have been demonstrated to increase the risk [11,12,13].

Invasive fungal infections (IFIs) are related to the impairment of cell-mediated immunity and are considered one of the major factors that constrain a successful outcome in patients receiving allogeneic SCT. During the pre-engraftment neutropenic phase, most fungal infections are due to yeasts (e.g., *Candida* spp.) and are related to neutropenia, severe mucositis, and central venous catheter use [10]. Although invasive mold infections are less frequent during this period, a history of multiple prior chemotherapies, iron overload, prior history of *Aspergillus* spp. infection, graft failure, and delayed engraftment increase the risk of aspergillosis [14,15]. On the other hand, the incidence of invasive aspergillosis is much greater during the post-engraftment period, with GVHD and prolonged corticosteroid use (>1 mg/kg per day of methylprednisolone or equivalents) being the most important risk factors [14]. *Mucormycosis* species are the second most common cause of mold infections in HCT recipients, whereas cases of *Fusarium* spp. and *Scedosporium* spp.-related disease have also been reported [16,17,18,19,20]. The risk of IFIs is substantially lower in autologous HCT recipients and mainly concerns candida infections related to intensive conditioning regimens causing substantial mucosal injury and prolonged neutropenia [21,22]. Invasive aspergillosis or other mold infections are not common in autologous HCT recipients unless they have a prior history of such infections.

The incidence of IFIs in adult patients with hematological malignancies treated with CAR T is globally between 5% and 10% [11,13]. Infection by both filamentous and non-filamentous fungi has been reported, and risk factors for invasive mold infection include ≥4 prior treatment lines, neutropenia below 500/mm^3^ prior to CAR T infusion, a dose of CAR T lymphocytes greater than 2 × 10^7^/kg, previous IFI, administration of tocilizumab and/or steroids [23].

Recently, a revised and updated version of the consensus definitions of IFDs was published by EORTC/MSGERC [24]. The consensus definitions are applicable to patients with cancer or recipients of stem cell or solid organ transplants and have been of great value to investigators who conduct clinical trials of antifungals, assess diagnostic tests, and perform epidemiologic studies. According to the updated revision, the definition of “probable” has been expanded and the scope of the category “possible” has been diminished. A fungal infection is considered proven based on the clinical characteristics of patients and microscopic analysis, culture, blood culture, serology tests, or tissue nucleic acid diagnosis where applicable. Probable IFD is diagnosed by the presence of at least one host factor, a clinical characteristic, and a piece of mycologic evidence. Possible IFDs are considered when a patient presents with a host factor and a clinical characteristic only. Updated guidelines for antifungal prophylaxis have also been proposed considering the paradigm shift in the treatment of hematologic malignancies in recent years and the development of novel antifungal agents [25,26,27,28]. Nevertheless, an evidence gap remains around the patient population that would benefit from antifungal prophylaxis and around the choice of antifungal agent for prophylaxis.

In data reported herein, the aim was to evaluate the epidemiology of IFIs following cellular therapy in our center, as well as to delineate risk factors that contribute to the development of IFIs in this patient population.

## 2. Materials and Methods

### 2.1. Cohort and Study Design

This is a retrospective single-center study conducted at the Hematology Clinic of George Papanicolaou Hospital. The aim of the study was to investigate the possible relationship between several factors associated with IFD and HCT. The study design was based on the methodology of previously published observational studies that were performed by our team [29,30,31]. The study concerned patients who underwent hematopoietic cell transplantation (HCT), CAR T cell therapy, or gene therapy with betibeglogene autotemcel during the years 2013 to 2022. (IFD). Indications for HCT were acute myeloid leukemia (AML), acute lymphoblastic leukemia (ALL), myelodysplastic syndromes (MDS), chronic myeloid leukemia (CML), primary myelofibrosis (PMF), multiple myeloma (MM), and lymphoma. Indications for CAR T cell therapy included relapsed/refractory lymphoma, and for gene therapy, adults with beta-thalassemia who required regular red blood cell transfusions were eligible. All the patients enrolled in the study were above 18 years old. To determine which patients to include, both the consensus definitions of IFD and guidance on the imaging of IFD were used [24,32]. Patient age, hematological disease, donor status, HLA matching with the donor, and the conditioning of the graft were analyzed. Alternative donors are considered mismatched unrelated, haploidentical donors and cord blood donors. Moreover, treatment with immunosuppressants, co-existence of a viral or bacterial infection, presence of a relapse, overall survival (OS), hematopoietic cell transplantation-specific comorbidity index (HCT-CI), presence of acute (grade II-III) and moderate or severe chronic graft-versus-host disease (GVHD) were evaluated. HCT-CI was used to calculate the transplant risk for each patient [33]. The diagnosis of acute and chronic GVHD was made based on clinical findings and/or biopsies according to standard criteria. Classic Glucksberg criteria were used for the calculation of the severity of acute GVHD, and we used the criteria of National Institutes of Health to assess the severity of chronic GVHD [34,35]. In Table 1, the baseline characteristics of the alloHCT recipients can be found. Primary antifungal prophylaxis in autologous and CAR T therapy was intravenous itraconazole during hospitalization and oral posaconazole during the outpatient period. All alloHCT recipients were assigned to primary antifungal prophylaxis with caspofungin and secondary prophylaxis with amphotericin. Following neutrophil engraftment until the cessation of immunosuppressive treatment, the primary prophylaxis switched to posaconazole and the secondary switched to isavuconazole. Furthermore, antifungal prophylaxis, type according to the current consensus, site of the IFD, the results of galactomannan and β-d-glycan tests, the culture result, the treatment that was used to treat the IFD and its outcome, treatment-related mortality (TRM) and results of imaging tests such as computed tomography (CT) and magnetic resonance imaging (MRI) were analyzed in order to identify a possible interaction between them and the severity of the disease. Galactomannan testing, and high-resolution computed tomography (HRCT) of the lung were routinely performed in each patient with suspicion of IFD (e.g., persistent fever unresponsive to broad-spectrum antibiotics). Further evaluation by bronchoscopy and bronchoalveolar lavage (BAL) was performed in case of imaging suggestive of IFI, whenever feasible. Additional tests (e.g., CT scan, MRI, culture) were performed based on clinical suspicion. Specifically, after the incubation of blood cultures, the species of all fungi that grew in the colonies were identified. For those considered possibly clinically relevant (clinical correlation, major and clear fungal development, specific species of fungi), a complete sensitivity check was performed. Co-infections were defined in symptomatic patients, with cultures in different samples to those of fungal infections. Bacteria from stools were also taken from symptomatic patients and not from screening.

### 2.2. Statistics

All statistical analyses were performed using SPSS 22.0 (IBM SPSS Statistics for Windows, Version 22.0. Armonk, NY, USA: IBM Corp). A descriptive analysis of all variables was performed using the median and range for continuous variables and the frequency for categorical variables. We assessed continuous variables for normality and performed a comparison using a *t*-test or Mann–Whitney test. The chi-square test was used for the comparison of categorical variables. In summary, we analyzed the following factors: age, disease, donor, HLA matching, conditioning, graft, severe acute and moderate/severe chronic GVHD, antifungal prophylaxis, immunosuppressants, IFD type (according to current consensus), galactomannan/β-d-glucan, cultures, imaging, bacterial/viral infections, IFD treatment, relapse, TRM, OS. The probabilities of OS were calculated using the Kaplan–Meier method, and survival curves were compared using a log-rank test. Furthermore, Cox regression analysis was performed for univariate and multivariate predictors of survival, with time-dependent covariates computed through SPSS analysis. The EZR software (Saitama, Japan) was used for the cumulative incidence (CI) of competing events’ analysis (http://www.jichi.ac.jp/saitama-sct/SaitamaHP.files/statmed.html (accessed on 1 June 2022)) [36]. The evaluation of statistical significance was performed using the Gray test and Fine and Gray regression modeling. A *p*-value ≤ 0.05 was considered to be significant.

## 3. Results

### 3.1. Study Population

Of the 950 patients that received cellular therapies during the study period, 19 received CAR T cell therapy due to relapsed/refractory lymphoma and 2 patients with thalassemia major received gene therapy, whereas 456 received autologous HCT and 473 underwent an allogeneic HCT according to EBMT guidelines at each period of treatment. None of the patients that received CAR T cell therapy or gene therapy developed an IFD. Three of the four hundred and fifty-six patients that received autologous transplantation developed a possible IFD; all of them had already been diagnosed with refractory or relapsed lymphomas (Appendix A). The study further analyzed the IFD in patients who received alloHCT.

Overall, 52 patients were suspected to have IFD after allogeneic HCT: 26 with AML, 18 with ALL, 4 with MDS, 2 with CML, 1 patient with NHL, and 1 patient with PMF. Ten (19.2%) patients received a graft from a matched sibling donor, 31 (59.6%) from a matched unrelated donor, and 11 (21.2%) from an alternative donor. The stem cell source was peripheral blood in 40 (76.9%) patients. More than two-thirds of patients underwent HCT in complete remission (22/52, 42.3% in first CR, 15/52, 28.8% in second CR, 4/52, 7.7% in third CR), and 11/52 (21.2%) in active disease.

### 3.2. Antifungal Prophylaxis and Patient Evaluation

The patients who undergo allogeneic HCT experience a neutropenic pre-engraftment phase which is associated with an increased risk of IFD, and the risk is further increased during the GVHD period and related immunosuppression. In our cohort, and in accordance with institutional guidelines, 41 of 52 patients received caspofungin as primary prophylaxis, and 11 of 52 patients received amphotericin as secondary prophylaxis until neutrophil engraftment. During the subsequent period of immunosuppression, caspofungin was replaced by posaconazole and amphotericin was replaced by isavuconazole.

### 3.3. Incidence and Characteristics of IFD

As far as concerns the patients that received alloHCT, 10 of 473 had been diagnosed with a proven IFD at a median of 226 (25–1146) days after receiving a transplant. A total of 11 out of 473 patients that underwent alloHCT had a probable infection at a median of 154 (12–469) post-transplant days, as shown in Table 2. In 31 of the 473 patients, a possible infection was suspected at a median of 112 (7–1353) days after the transplant (Table 2). The median age of the patients was 34 years, and there was no difference between the other groups. Some patients that received alloHCT and were diagnosed with an IFD presented a concurrent bacterial or viral infection. Specifically, the coexistence of a bacterial infection was identified in 20 out of 52 patients and of a viral infection in 17 of 52 patients (Table 3). Three of fifty-two patients were diagnosed with a second IFD, one patient with probable and two with a proven IFD; two of them relapsed and one succumbed to TRM. Various types of fungi were identified in patients. Specifically, candida species were isolated in nine patients; *Fusarium* spp. and *Aspergillus* spp. were isolated in two (Table 4). Patients with a positive galactomannan test had *Aspergillus* species. In addition, two of the patients with IFD were diagnosed with mucormycosis.

### 3.4. Independent Risk Factors for IFD

The multivariate analysis revealed independent risk factors for IFD: type of donor (IFD present in 20% of alternative, 12% of unrelated, and 5% of sibling donors, *p* = 0.006) and moderate/severe chronic GVHD (IFD present in 15% of moderate/severe versus 8% of mild GVHD, *p* = 0.029). 

### 3.5. Overall Survival (OS)

In the multivariate analysis, IFD had a negative impact on the OS of the patients (*p* = 0.044) regardless of the type of donor and the existence of chronic GVHD, as shown in Figure 1. The four-year OS was significantly lower in those with proven IFD compared to those who developed a probable IFD. Specifically, patients with a proven infection had a 27.4% possibility of survival in the first four years after transplantation whereas those with probable IFD had a 75.4% survival possibility (*p* = 0.041). 

## 4. Discussion

In the present study, we evaluated the epidemiology and prognostic factors of IFIs in patients who received HCT and other cellular therapies at our center according to standard operating procedures over the last decade.

According to our findings, none of the patients who underwent CAR T cell therapy developed IFD. As reported in previous studies, the overall incidence of IFDs after CAR T cell therapy is low and it is mostly associated with prior HCT [13,37,38]. However, Cordeiro et al. showed that 5 out of 54 patients (9%) developed fungal infections in a period longer than 90 days after treatment, but the majority of them were mild and able to be managed in outpatient settings [39]. Given that, the administration of antifungal prophylaxis needs to be personalized for patients undergoing CAR T cell therapy [28]. Furthermore, zero prevalence of IFDs in patients treated with gene therapy was reported. However, our sample size was small. No other studies examining the prevalence and risk of fungal infections after gene therapy were identified.

In addition, a very low incidence of IFD in patients who had autologous HCT was reported, and those affected suffered from primary refractory/relapsed lymphomas. According to a consistent number of studies, the IFI rate is significantly low among autologous stem cell transplant (HSCT) recipients, 0.4–1.5%, and so is the mortality rate [18,40,41,42,43]. In accordance with our findings, Srinivasan et al. concluded that fungal infections among children and adolescents who received autologous HCT are not common [44]. There is some evidence that the prevalence of IFI is higher in patients who receive autologous grafts consisting of modified (CD34-selected) stem cells [45]. These epidemiological data have an impact on choosing the appropriate antifungal prophylaxis for patients receiving autologous HCT.

A total of 52 of the allogeneic HCT recipients were observed to develop IFD (31 possible, 11 probable, and 10 proven). A multicenter prospective observational study found that the incidence of IFI is 9.0% in allogeneic hematopoietic stem cell transplantation [46]. Chu et al. reported that the prevalence of IFI may be up to 5–8% [47]. In our study, the median post-transplant day of IFD was 112, 154, and 226 for patients with possible, probable, and proven IFI, respectively. In accordance with our findings, a retrospective review of 271 subjects undergoing allogeneic HCT, found that 67% of IFIs occurred more than 100 days after HCT, after full hematologic recovery had occurred, and all patients received antifungal prophylaxis [48]. Another single-center study conducted in Lebanon showed that 67/195 allo-HCT recipients, who received voriconazole as primary antifungal prophylaxis, developed IFD (proven, probable, and possible) [49].

The development of IFD after the normalization of neutrophil count may be related to the use of immunosuppressant agents against chronic GVHD and GVHD itself [48,50]. The incidence of IFI in the pediatric population after allogeneic HCT is comparable to the rate that is reported here for adult patients, but most of the events occurred within the first month after transplantation [51]. We underline that IFI occurred comparatively late after HCT in patients with completed engraftment, in comparison to the pediatric data reported, in which infections occurred during neutropenia. Unfortunately, CD4+ T cell regeneration and individualized sequences of treatments are not available in this analysis but could be an interesting avenue in future work, in order to investigate whether the late incidence is possibly associated with these factors. It is interesting that there was a predominance of candida species infections in patients who received allogeneic HCT. Sakellari et al. identified 7 patients among 108 allogeneic HCT recipients who developed candidemia and they noted that infection by candida species is associated with impaired cell-mediated immunity, which is the result of immunosuppression [52]. It is well established that the use of fluconazole as a prophylactic agent has resulted in a decrease in cases of invasive candidiasis, leading to an outbreak of invasive mold infections (IMIs), especially IA [53]. In our cohort, allogeneic HCT recipients who then developed invasive fungal disease were previously assigned to primary antifungal prophylaxis with caspofungin and secondary prophylaxis with amphotericin, and then switched to posaconazole and isavuconazole, respectively, following neutrophil engraftment. Harrison et al. reported that the use of antifungal prophylaxis decreased the 1-year incidence for IFI [54]. Given the availability of new oral antifungal prophylaxis, an interesting future study would be an analysis of the absolute numbers of allogeneic HCT performed in our center during the study period and reported IFI, presented by calendar year, in relation to the antifungal agents that patients were assigned. In conclusion, the development of fungal infections in immunosuppressed patients who already receive antifungal agents demonstrates an urgent need to develop more effective therapeutic strategies.

Our study has several limitations that should be recognized due to its design. The current study is a single-center study and is retrospective in nature. Accordingly, the epidemiology and the conclusions of our study might not be representative of the spectrum of fungal infections distributed across various transplant centers and reflects only the epidemiology from Northern Greece. A prospective design or larger number of patients included would eliminate the possible biases of a retrospective study. In addition, the power of our study does not allow us to point out significant differences between patients with infections from different species. The number of patients who received CAR T cell therapy or gene therapy was relatively low, and these findings should be verified by additional studies in this field. Furthermore, acute/chronic GVHD were introduced into the model as traditional variables.

In this study, risk factors for IFD in patients receiving allogeneic HSCT included the type of donor and moderate/severe chronic GVHD [45,46,48]. A series of studies correlated the HLA-matched unrelated donors with an increase in IFIs. Mikulska et al. showed that GVHD therapy resulted in an important increase in late IFI in the patients of their study [50]. As shown, GVHD might be the most relevant factor to the outcome of these patients. These findings suggest that the integrity of the immune system of the patient is crucial to control fungal infections, thus systematic monitoring and antifungal prophylaxis in these patients is crucial. The 4-year overall survival rate for IFI was significantly different for subjects without IFI (*p* = 0.041). In accordance with previous studies, the overall mortality is markedly higher in patients with IFD [46,55,56].

## 5. Conclusions

In summary, our study acknowledges the low prevalence of proven IFD in patients who receive novel antifungal prophylaxis. However, IFD is associated with poor out-comes in patients who receive alloHCT. According to our findings, possible and probable IFD assessed by host and clinical factors have been shown to have similar outcomes. Thus, a more extensive diagnostic work-up (e.g., early CT scan and BAL) combined with a broad application of sensitive mycological testing is crucial for high-risk patients, including alternative alloHCT recipients and patients with chronic GVHD. Further studies are required to provide more sensitive diagnostic strategies for IFD.

## Figures and Tables

**Figure 1 cancers-15-03529-f001:**
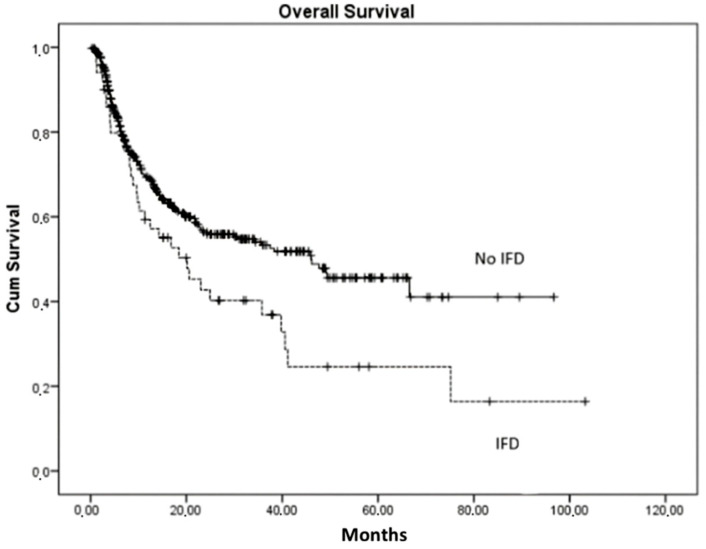
Impact of IFD post-alloHCT on overall survival (OS). Any type of ΙFD associates with lower OS.

**Table 1 cancers-15-03529-t001:** Baseline characteristics of allogeneic HCT recipients.

Characteristic	AlloHCT with IFD (n = 52)	AlloHCT Recipients without IFD (n = 421)	*p*-Value
**Hematological disease, n**			0.324
AML	26	216	
ALL	18	87	
MDS	4	32	
CML	2	25	
NHL	1	32	
PMF	1	19	
Other		10	
**Type of donor, n**			0.012
Graft from a matched sibling donor	10	232	
Graft from a matched unrelated donor	31	164	
Graft from an alternative donor	11	25	
**HCT, n**			0.127
HCT in complete remission in CR	41	369	
HCT in active disease	11	52	

HCT, Hematopoietic cell transplantation; IFD, Invasive fungal disease; AML, Acute myeloid leukemia; ALL, Acute lymphoblastic leukemia; MDS, Myelodysplastic syndrome; CML, Chronic myeloid leukemia; NHL, Non-Hodgkin lymphoma, PMF; Primary myelofibrosis; CR, Complete remission.

**Table 2 cancers-15-03529-t002:** IFD in allogeneic HCT recipients.

IFD	Possible	Probable	Proven
Patients (n)	31	11	10
Median post-transplantation day	112	154	226
Positive Galactomannan (n)	0	5	3
Positive cultures (n)	0	6	7
Histopathological evidence	0	0	3
Second IFD episode (n)	0	1	2

IFD, Invasive fungal disease; HCT, Hematopoietic cell transplantation.

**Table 3 cancers-15-03529-t003:** Detailed report of bacterial pathogens and viruses detected in patients with IFD.

Bacterial Infections	Patients (n)	Viral Infections	Patients (n)
Blood		Blood CMV EBV Influenza A	8 9 1
*Staphyloccocus hominis*	1		
*Staphylococcus haemolyticus*	1		
*Enteroccus faecium*	1		
*Pseudomonas aeruginosa*	2		
*Klebsiella pneumoniae*	2		
*Eserichia coli*	1		
*Sphingomonas paucimobilis*	1		
Sputum			
*Enterococcus faecium*	1		
*Klebsiella pneumoniae*	3		
*Pseudomonas aeruginosa*	3		
*Pneumocystis jirovecii*	1		
Stools			
*Klebsiella pneumoniae*	2		
Urine			
*Klebsiella pneumoniae*	1		
	20		17 *

CMV, cytomegalovirus; EBV, Epstein-Barr virus; * Coexistence of CMV and EBV viremia was identified in one patient.

**Table 4 cancers-15-03529-t004:** Fungal infections in patients with IFD.

Fungal Infections	Patients (n)
*Candida* spp.	9
*Fusarium*	2
*Aspergillus*	2
*Mucormycosis*	2

IFD, Invasive fungal disease.

## Data Availability

The authors declare that the data supporting the findings of this study are available within the paper.

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
