# Peer review of "Risk Factors, Prevalence, and Outcomes of Invasive Fungal Disease Post Hematopoietic Cell Transplantation and Cellular Therapies: A Retrospective Monocenter Real-Life Analysis"

_cancers, 2023, doi:10.3390/cancers15133529_

Round 1
Reviewer 1 Report
This research is under the scope of this journal; the topic is relevant for readers, and this research deals with potentially significant knowledge to the field. And It will be important of Cancer knowledge. The topic is relevant for readers and this study deals with potentially significant knowledge to the field and open new way for future studies.
Please add more keywords, and order the keywords / Mesh terms alphabetically
Please consider explain better limitations.
Please consider other references about allogenic Cell Therapy: PMID: PMID: 32486396
Sample size calculation is not clear. Please better describe the primary outcome utilized, standard deviation and the mean average among groups.
Check references
Please, add more information in figure legends
Author Response
Author’s reply
Reviewer 1
Dear Reviewer,
This research is under the scope of this journal; the topic is relevant for readers, and this research deals with potentially significant knowledge to the field. And It will be important of Cancer knowledge. The topic is relevant for readers and this study deals with potentially significant knowledge to the field and open new way for future studies.
Reply: We would like to thank Reviewer 1 for all his/her evaluating our work and highlighting it as a valuable source of information for invasive fungal disease in recipients of cellular therapy.
Please add more keywords, and order the keywords / Mesh terms alphabetically.
Reply: More keywords were added and ordered alphabetically.
Please consider explain better limitations.
Reply: We expanded the limitations paragraph in order to explain them better.
Please consider other references about allogeneic Cell Therapy: PMID: PMID: 32486396
Reply: Indeed we added this interesting reference in our manuscript.
Sample size calculation is not clear. Please better describe the primary outcome utilized, standard deviation and the mean average among groups.
Reply: We thank the Reviewer for this comment. Our aim was to provide real-life data with the primary outcome being the prevalence of fungal infections in our large cohort.
Check references
Reply: Thank you for pointing this out. References were checked.
Please, add more information in figure legends
Reply: More information was added in figure legend.

Reviewer 2 Report
The manuscript presented by Gavriilaki et al. shows important results of a 10-year prospective study in a reference hospital. Overall, the manuscript is well-written but many typo errors were seen in the document (probably due to text formatting issues). I have some minor questions / doubts & suggestions for authors:
1) Risk factors. Throughout the text authors have cited risk factors related to invasive fungal infections in patients who received cellular therapies. I wonder if the studied cohort may have any risk factors besides those presented (type of donor and moderate/severe chronic GVHD). Please show all the risk factors studied and highlight those significant.
2) Regarding antifungal prophylaxis. This information was not clear to me: all patients underwent the same antifungal prophylactic scheme?
3) In general, the number of patients with IFD (possible, probable or proven) was very small. I wonder if authors could show data regarding each of them in a more detailed way. Mainly for proven IFI group (for example: a) Did patients positive for galactomannan test were infected by which fungus? b) Patients infected by Candida showed co-infections with which bacteria species? What about the age of these patients? Proven IFI occurred after how many days after transplant? etc)
4) Definitions. Authors must explain for the reader about the terms “possible”, “probable” and “proven”. Which criteria were used to classify IFI into one of these categories? Please define “co-infections”: were they searched in symptomatic patients? If so, did they occur concomitantly to fungal isolation (in the same blood sample)? Bacteria recovered from stools were isolated from patients with diarrhea or were swab-screening test for monitoring colonizing patients?
5) Microbiology. Please cite the methods employed for fungal, bacterial and virus identification. As Mycologist, I understand how difficult it is to achieve species-level identification for many pathogens… but, in general, the majority of facilities available in reference institutions are able to identificate Candida spp. thoroughly. Did authors find any information regarding antibiotic susceptibility of bacterial isolated from blood? Antibiotic resistance is an actual menace to vulnerable patients.
Author Response
Reviewer 2
Dear Reviewer,
The manuscript presented by Gavriilaki et al. shows important results of a 10-year prospective study in a reference hospital. Overall, the manuscript is well-written but many typo errors were seen in the document (probably due to text formatting issues).
We would like to thank Reviewer 2 for all his/her comments and the valuable feedback.
I have some minor questions / doubts & suggestions for authors:
1) Risk factors. Throughout the text authors have cited risk factors related to invasive fungal infections in patients who received cellular therapies. I wonder if the studied cohort may have any risk factors besides those presented (type of donor and moderate/severe chronic GVHD). Please show all the risk factors studied and highlight those significant.
Reply: We thank the Reviewer for this comment. All factors studied have been added in summary in the statistical analysis.
2) Regarding antifungal prophylaxis. This information was not clear to me: all patients underwent the same antifungal prophylactic scheme?
Reply: Thank you for pointing this out. We modified our manuscript.
3) In general, the number of patients with IFD (possible, probable or proven) was very small. I wonder if authors could show data regarding each of them in a more detailed way. Mainly for proven IFI group (for example: a) Did patients positive for galactomannan test were infected by which fungus? b) Patients infected by Candida showed co-infections with which bacteria species? What about the age of these patients? Proven IFI occurred after how many days after transplant? etc)
Reply: We agree with your comment, and we incorporated your suggestions to our manuscript. Patients with galactomannan positive had Aspergillus species. Median age was 34 years, no difference between the other groups. Days after transplant were originally stated in our manuscript.
4) Definitions. Authors must explain for the reader about the terms “possible”, “probable” and “proven”. Which criteria were used to classify IFI into one of these categories? Please define “co-infections”: were they searched in symptomatic patients? If so, did they occur concomitantly to fungal isolation (in the same blood sample)? Bacteria recovered from stools were isolated from patients with diarrhea or were swab-screening test for monitoring colonizing patients?
Reply: The reviewer is right. Our perspective has been more from the clinical point of view, but the perspective of microbiology is also important. Co-infections were defined in symptomatic patients, with cultures in different samples than those of fungal infections. Bacteria from stools were also from symptomatic patients and not from screening.
5) Microbiology. Please cite the methods employed for fungal, bacterial and virus identification. As Mycologist, I understand how difficult it is to achieve species-level identification for many pathogens… but, in general, the majority of facilities available in reference institutions are able to identificate Candida spp. thoroughly. Did authors find any information regarding antibiotic susceptibility of bacterial isolated from blood? Antibiotic resistance is an actual menace to vulnerable patients.
Reply: We would like to thank for your suggestion. We made the essential amendments in the methods section.

Reviewer 3 Report
In the present manuscript, Gavriilaki et al. report retrospective data on invasive fungal infections in patients with hematologic disorders (predominantly malignancies) receiving cellular therapies at a major southern European center. The reported study population is substantial and from the recent decade, which makes it relevant as real-world report of cellular therapy complications in southern Europe. The methodology is overall sound. However, there are some minor aspects that should be addressed to improve the present manuscript:
Minor:
- Please do provide detailed results on the 3 patients with autologous HCT and invasive fungal infection as supplement table.
- Please further perform a descriptive analysis of the allo patients with at least suspected fungal infection for the calendar year of transplantation. Given the availability of new oral prophylaxis means with e.g. posaconazole as pills instead of liquid posaconazole, this analysis could provide insights on more recent changes in prevalence.
- Single center studies may shed light on regional differences in practice and disease prevalence. I would like to see the results even more contextualized with similar reports from other centers to have a broader view of the patient outcomes in different parts of Europe and overseas.
- Please briefly detail in the methods section the variables that were used for multivariate analysis.
- Please add to the limitations section that the number of CART patients may be too low to detect fungal infections given the prevalence in previous studies. Here, the prevalence has been lower than 1/19.
- Please taper down the statement in the legend of Figure 1. Better write “associates with” lower OS, as the fungal infection is a time-dependent event occurring after HCT.
- The written English is overall fine. In the first sentence of the simple summary better write “IFD are a significant cause…” or “contribute significantly to…”
- The baseline table 1 should better match with the methods cohort section.
- Suggestion. For some of the presented analyses, the patients with “proven” and “probable” fungal infection could also be analyzed “together”, maybe you can provide it in the supplement.
- Suggestion. I think it is relevant to underline that the reported infections occurred comparatively late after HCT in patients with completed engraftment, in particular as compared to the pediatric data which reported infections during neutropenia; and there is a predominance of candida species, which is also interesting. In parallel the outcomes of these patients are better than I would have expected, as the 12 months OS did not differ between patients with and without invasive fungal infections. Only one patient died due to the infection, if I read it right. The curves separate later indicating that most patients were successfully treated. The multivariate analysis gives already a hint that GVHD may be the most relevant factor in this context. Did the late incidence possibly associate with the termination of antifungal prophylaxis after CD4+ T cell regeneration? Also the duration of treatment and sequence of antifungals is of potential interest and could be addressed for the 10 to 21 patients of concern.
Author Response
Reviewer 3
Author’s reply
Dear Reviewer,
We would like to thank the Reviewer 3 for all the insightful comments that were crucial for the improvement of our manuscript.
In the present manuscript, Gavriilaki et al. report retrospective data on invasive fungal infections in patients with hematologic disorders (predominantly malignancies) receiving cellular therapies at a major southern European center. The reported study population is substantial and from the recent decade, which makes it relevant as real-world report of cellular therapy complications in southern Europe. The methodology is overall sound. However, there are some minor aspects that should be addressed to improve the present manuscript:
We would like to thank Reviewer 3 for all his/her interest in evaluating our manuscript. Indeed his/her comments were very meaningful for our work.
Minor:
- Please do provide detailed results on the 3 patients with autologous HCT and invasive fungal infection as supplement table.
Reply: As per your recommendation, we have provided detailed results on the 3 patients with autologous HCT who developed a possible IFD.
- Please further perform a descriptive analysis of the allo patients with at least suspected fungal infection for the calendar year of transplantation. Given the availability of new oral prophylaxis means with e.g. posaconazole as pills instead of liquid posaconazole, this analysis could provide insights on more recent changes in prevalence.
Reply: We would like to thank you for your suggestion. It is something that we pursue for the future, and we have added it as such in the revised manuscript. Some limitations of this analysis would include the very low numbers of infections each year and the fact that for some years posaconazole as pills was used simultaneously to liquid posaconazole.
- Single center studies may shed light on regional differences in practice and disease prevalence. I would like to see the results even more contextualized with similar reports from other centers to have a broader view of the patient outcomes in different parts of Europe and overseas.
Reply: Indeed, that would be very interesting. In the revised discussion section, you can find the comparison of our findings with the data reported in other single center studies held overseas.
- Please briefly detail in the methods section the variables that were used for multivariate analysis.
Reply: We made the changes that you asked in the statistics analysis section.
- Please add to the limitations section that the number of CART patients may be too low to detect fungal infections given the prevalence in previous studies. Here, the prevalence has been lower than 1/19.
Reply: We agree with this comment. The essential changes were made in the limitations section in order to underline the relatively low number of CAR-T patients included in our cohort.
- Please taper down the statement in the legend of Figure 1. Better write “associates with” lower OS, as the fungal infection is a time-dependent event occurring after HCT.
Reply: As other reviewers suggested we amended the statement in the legend of figure 1.
- The written English is overall fine. In the first sentence of the simple summary better write “IFD are a significant cause…” or “contribute significantly to…”
Reply: Thank you for this suggestion.
- The baseline table 1 should better match with the methods cohort section.
Reply: The table 1 is now part of the methods cohort section.
- Suggestion. For some of the presented analyses, the patients with “proven” and “probable”
fungal infection could also be analyzed “together”, maybe you can provide it in the supplement.
Reply: We would like to thank the reviewer for his/ her suggestion. However, we added some more data about the proven IFD in the results section in order to make our analysis more detailed.
- Suggestion. I think it is relevant to underline that the reported infections occurred comparatively late after HCT in patients with completed engraftment, in particular as compared to the pediatric data which reported infections during neutropenia; and there is a predominance of candida species, which is also interesting. In parallel the outcomes of these patients are better than I would have expected, as the 12 months OS did not differ between patients with and without invasive fungal infections. Only one patient died due to the infection, if I read it right. The curves separate later indicating that most patients were successfully treated. The multivariate analysis gives already a hint that GVHD may be the most relevant factor in this context. Did the late incidence possibly associate with the termination of antifungal prophylaxis after CD4+ T cell regeneration? Also the duration of treatment and sequence of antifungals is of potential interest and could be addressed for the 10 to 21 patients of concern.
Reply: We really appreciate this in-depth expert analysis. We have added some of these advantages to our study in the discussion. Indeed, we agree that GVHD might be the most relevant factor. Unfortunately, CD4 T cell regeneration or individualized sequence of treatments are not available in this analysis but could have been work in the future.

Reviewer 4 Report
Risk Factors, Prevalence, and Outcomes of Invasive Fungal Disease Post Cellular Therapies: "a retrospective monocenter real-3 life analysis
Authors analized IFD in patients receiving CAR-T (19) gene therapy (2) autologous HCT (456) allogeneic HCT (473)
The title is a bit misleading as most patients were recipients of HCT with only few CAR-T and other more “modern” cellular treatment.
Antifungal prophylaxis with caspofungin in allo HCT recipients, Ampho B was used for secondary prophylaxis followed by posaconazole OR? isavuconazole respectively
What was the antifungal prophylaxis in recipients of autologous HCT and of CAR-T?
As none of the CAR-T patients developed IFD and 3 of the autologous HCT receipients had possible IFD the in depth analysis pertains to allogeneic HCT recipients only.
The study is too small to analyzed IFD in CAR-T recipients of in recipients of gene therapy protocols.
Details:
What is: outcomes of IFD cellular therapies? IFD in recipients of cellular therapies?
in or-der to identify
our JACIE-accredited: appears 3 times in the text, I assume the center is accredited = a singularity. therefore mentioning this fact once should suffice.
Acute/chronic GvHD as risk factor: were these terms introduced in the model as time varying?
Author Response
Reviewer 4
Author’s reply
Dear Reviewer,
We would like to thank Reviewer 4 for all the comments that helped us to revise
and improve this manuscript.
Risk Factors, Prevalence, and Outcomes of Invasive Fungal Disease Post Cellular Therapies: "a retrospective monocenter real-3 life analysis
Authors analyzed IFD in patients receiving CAR-T (19) gene therapy (2) autologous HCT (456) allogeneic HCT (473)
The title is a bit misleading as most patients were recipients of HCT with only few CAR-T and other more “modern” cellular treatment.
Reply: The title has now been amended to “Risk Factors, Prevalence, and Outcomes of Invasive Fungal Disease Post Hematopoietic Cell Transplantation and Cellular Therapies: "a retrospective monocenter real-life analysis""”.
Antifungal prophylaxis with caspofungin in allo HCT recipients, Ampho B was used for secondary prophylaxis followed by posaconazole OR? isavuconazole respectively
Reply: We would like to thank you for this question. Amphotericin B, used for secondary prophylaxis, was followed by isavuconazole. We made the essential amendments in the methods section to make it more obvious.
What was the antifungal prophylaxis in recipients of autologous HCT and of CAR-T?
Reply: We thank the Reviewer for this comment. Primary antifungal prophylaxis in autologous and CAR-T was intravenous itraconazole during hospitalization and oral posaconazole during the outpatient period.
As none of the CAR-T patients developed IFD and 3 of the autologous HCT receipients had possible IFD the in depth analysis pertains to allogeneic HCT recipients only.
The study is too small to analyzed IFD in CAR-T recipients of in recipients of gene therapy protocols.
Reply: We agree with the Reviewer’s comments, and we addressed this point in the limitations paragraph of the discussion section.
Details:
What is: outcomes of IFD cellular therapies? IFD in recipients of cellular therapies?
Reply: We thank the Reviewer for this comment. Essential changes were made in the abstract.
in or-der to identify
our JACIE-accredited: appears 3 times in the text, I assume the center is accredited = a singularity. therefore mentioning this fact once should suffice.
Reply: We thank the Reviewer for this comment. We have updated the manuscript
accordingly.
Acute/chronic GvHD as risk factor: were these terms introduced in the model as time varying?
Reply: We appreciate this comment. Acute/chronic GvHD were introduced in the model as traditional variables. We have added this as a limitation of the present study.
